

# Modular invariance faces precision neutrino data

## Juan Carlos Criado[1] and Ferruccio Feruglio[2]

**1** CAFPE and Departamento de Física Teórica y del Cosmos, Universidad de Granada,
Campus de Fuentenueva, E-18071, Granada, Spain
**2** Dipartimento di Fisica e Astronomia 'G. Galilei', Università di Padova,
INFN, Sezione di Padova, Via Marzolo 8, I-35131 Padua, Italy

## Abstract

We analyze a modular invariant model of lepton masses, with neutrino masses originating either from the Weinberg operator or from the seesaw. The constraint provided by modular invariance is so strong that neutrino mass ratios, lepton mixing angles and Dirac/Majorana phases do not depend on any Lagrangian parameter. They only depend on the vacuum of the theory, parametrized in terms of a complex modulus and a real field. Thus eight measurable quantities are described by the three vacuum parameters, whose optimization provides an excellent fit to data for the Weinberg operator and a good fit for the seesaw case. Neutrino masses from the Weinberg operator (seesaw) have inverted (normal) ordering. Several sources of potential corrections, such as higher dimensional operators, renormalization group evolution and supersymmetry breaking effects, are carefully discussed and shown not to affect the predictions under reasonable conditions.



# 1   Introduction

In the last few years neutrino physics has entered the era of precision, a goal that could not even be conceived when neutrino oscillations were discovered twenty years ago. The well-established three-neutrino framework and its general parametrization works very well for almost all experimental results. Squared mass differences and mixing angles are known to a few percent precision, while global fits [1–5] are cornering both the Dirac phase and the type of mass ordering. Despite this tremendous progress there is not yet a clear indication about a unique guiding principle providing an explanation of the present pattern of lepton masses and mixing angles in terms of few underlying parameters. A similar state of affairs can be recognized in the quark sector, which grand unification suggests to be strictly related to the leptonic one. Adequate theoretical tools supplying predictions matching the present level of accuracy could represent a significant step towards the solution of the flavour puzzle.

To date one of the few available frameworks where quantitative statements about lepton mass parameters can be formulated is the one of flavour symmetries [6–14]. Yet the variety of the offered possibilities is so wide that a sharp selection should be done to met the desired predicability requirements. As a matter of fact there is no hint of an exact flavour symmetry neither in the quark sector nor in the lepton one. Only broken symmetries have the chance of being realistic and this poses several technical problems. The breaking sector typically introduces many independent parameters. Models are often formulated in terms of effective field theories valid below a certain energy scale, and several independent higher-dimensional operators can significantly contribute to the same physical parameters. A complicated vacuum alignment is frequently required, with many flavons whose vacuum expectation values (VEV) should be conveniently oriented in generation space. All this, beyond being rather tricky from the technical viewpoint, goes to the detriment of predictability, to the point that anarchy [15–19] or its variants have often being invoked as an interpretation of the elusive neutrino features.

Supersymmetric models with modular invariance as flavour symmetry, recently proposed in ref. [20] and briefly summarized here, have a chance to provide the desired framework, alleviating some of the above mentioned difficulties. They are characterized by a small number of Lagrangian parameters. Moreover, a truly remarkable feature of these models is that all higher-dimensional operators in the superpotential are unambiguously determined in the limit of unbroken supersymmetry. Modular invariance arises in compactifications of the heterotic string on orbifolds [21–25], in D-brane compactifications [26–32], in magnetized extra-dimensions [33–35]. It has been investigated in connection with the flavour problem in ref. [36–46] and, more recently, in ref. [47,48]. The formalism of modular invariant supersymmetric field theories has been described in [49,50]. In this work we analyse one supersymmetric modular invariant model describing the lepton sector. Actually we consider two variants of the model where neutrino masses originate either from the Weinberg operator or from the

seesaw mechanism. At low-energy and in both variants, the superpotential depends only on one overall mass scale and 3 dimensionless parameters. The latter are adjusted to reproduce the charged lepton masses, which are not predicted, but just fitted as in the Standard Model. All the remaining 8 dimensionless parameters, neutrino mass ratios, lepton mixing angles, Dirac and Majorana phases, do not depend on any Lagrangian parameter. They do depend on the vacuum structure of the model, controlled by the VEVs of a modulus field and a flavon, minimally described by a total of 3 real parameters. The mechanism selecting the vacuum of the world we live in is still largely unknown, and we do not look for a solution of the vacuum problem in our models. We rather scan the VEVs treating them as free parameters. Therefore our models attempt to predict the 8 dimensionless neutrino parameters in terms of 3 effective real parameters, the symmetry breaking VEVs. This represents a change of perspective with respect to most of the existing models, where the vacuum is obtained through the solution of a dynamical problem such as the minimization of the energy density of the theory, while a remaining set of Lagrangian parameters is used to fit masses and mixing angles. As we will see, for both models, the attempt is successful and all the presently known mass combinations and mixing angles are correctly reproduced, which we consider a highly non-trivial result.

The two models are indeed very similar: in the first one neutrino masses come from the Weinberg operator. In the second one they come from the see-saw mechanism. The symmetry setup is the same and the difference is in the ordering of neutrino masses: inverted ordering for the Weinberg operator and normal ordering for the see-saw. In ref. [20] these models were studied in the limit where the charged lepton sector gives no contribution to the lepton mixing and the results were encouraging. With only 2 effective real parameters, the complex VEV of the modulus field, only the reactor angle $\theta_{13}$ was outside its allowed experimental range. In the present note we account for a possible contribution to the mixing from the charged lepton mass matrix, turning on a single additional real parameter, bringing all predictions in agreement with the data. Moreover we quantitatively discuss the effects due to the running of the mass/mixing parameters from the high scale, where they are presumably defined, down to the low energy scale where they are observed. Finally, we estimate the impact of supersymmetry breaking terms, required in any realistic framework accounting for the existing bounds on supersymmetric particles.

## 2 The Models

Here we are mainly interested in the Yukawa interactions, described by the following action:

$$\mathcal{S} = \int d^4x d^2\theta d^2\bar{\theta}\, K(\Phi, \bar{\Phi}) + \int d^4x d^2\theta\, w(\Phi) + h.c. \quad , \tag{1}$$

where $K(\Phi, \bar{\Phi})$, the Kähler potential, is a real gauge-invariant function of the chiral superfields $\Phi$ and their conjugates and $w(\Phi)$, the superpotential, is a holomorphic gauge-invariant function of the chiral superfields $\Phi$. The chiral superfields $\Phi = (\tau, \varphi^{(I)})$ include the modulus [1] $\tau$ and the remaining chiral supermultiplets, $\varphi^{(I)}$, transforming under the modular group $\Gamma$ as

$$\begin{cases} \tau & \rightarrow & \gamma\tau \equiv \dfrac{a\tau + b}{c\tau + d} \\[2mm] \varphi^{(I)} & \rightarrow & (c\tau + d)^{k_I}\rho^{(I)}(\gamma)\varphi^{(I)} \end{cases}, \tag{2}$$

---

[1]Also the notation $T = -i\tau$ is used in the literature. The modulus $\tau$ describes a dimensionless chiral supermultiplet, depending on both space-time and Grassmann coordinates.

with $a$, $b$, $c$ and $d$ integers satisfying $ad - bc = 1$ and $\rho^{(I)}$ a representation of the quotient group $\Gamma_3 = \Gamma/\Gamma(3)$ [2]. The real number $k_I$ is called the weight [3] of the multiplet $\varphi^{(I)}$.

We discuss two specific realizations of this framework [4], whose main difference is the origin of the Majorana masses for light neutrinos, either the Weinberg operator for Model 1, or the seesaw mechanism in Model 2. Their field content and its transformation properties are listed in table 1. Beyond the modulus $\tau$, Model 1 contains the $SU(2)_L$ lepton singlets $E_i^c$ ($i = 1, 2, 3$), three generations of $SU(2)_L$ doublets $L$, the Higgses $H_{u,d}$, a gauge invariant flavon $\varphi$. Model 2 has in addition three generations of gauge singlets $N^c$. In our conventions both the modulus $\tau$ and the flavon $\varphi$ are dimensionless fields. The correct dimensions can be recovered by an appropriate rescaling. Weights are specified in table 2.

Table 1: Chiral supermultiplets, transformation properties and weights. Model 1 has no gauge singlets $N^c$.

|  | $(E_1^c, E_2^c, E_3^c)$ | $N^c$ | $L$ | $H_d$ | $H_u$ | $\varphi$ |
|---|---|---|---|---|---|---|
| $SU(2)_L \times U(1)_Y$ | $(1, +1)$ | $(1, 0)$ | $(2, -1/2)$ | $(2, -1/2)$ | $(2, +1/2)$ | $(1, 0)$ |
| $\Gamma_3 \equiv A_4$ | $(1, 1'', 1')$ | $3$ | $3$ | $1$ | $1$ | $3$ |
| $k_I$ | $(k_{E_1}, k_{E_2}, k_{E_3})$ | $k_N$ | $k_L$ | $k_d$ | $k_u$ | $k_\varphi$ |

Table 2: Weights of chiral multiplets. Model 1 has no gauge singlets $N^c$.

|  | $k_{E_i}$ | $k_N$ | $k_L$ | $k_d$ | $k_u$ | $k_\varphi$ |
|---|---|---|---|---|---|---|
| Model 1 | $-2$ | $-$ | $-1$ | $0$ | $0$ | $+3$ |
| Model 2 | $-4$ | $-1$ | $+1$ | $0$ | $0$ | $+3$ |

We ask invariance of the action $\mathcal{S}$ under the transformations (2), which implies the invariance of the superpotential $w(\Phi)$ and the invariance of the Kähler potential up to a Kähler transformation [49, 50]:

$$\begin{cases} w(\Phi) \to w(\Phi) \\ K(\Phi, \bar{\Phi}) \to K(\Phi, \bar{\Phi}) + f(\Phi) + f(\overline{\Phi}) \end{cases} . \tag{3}$$

The modular invariance of the superpotential $w(\Phi)$ is guaranteed by two simple rules:

1. $w(\Phi)$ should be invariant under the group $\Gamma_3$, as in the usual setup of discrete flavour symmetries.

2. The superpotential $w(\Phi)$ should have vanishing total weight.

To enforce these rules it is convenient to observe that there are holomorphic functions $f_i(\tau)$ of the modulus $\tau$, called modular forms [51], with simple transformation properties under the

---

[2]Any principal congruence subgroup $\Gamma(N)$ of $\Gamma$ can be used in this formalism, the integer $N$ being called the level. In our work we choose $N = 3$. $\Gamma_3$ is isomorphic to $A_4$.

[3]Notice the different sign convention with respect to ref. [20].

[4]They correspond to Examples 1 and 2 of ref. [20].

modular group, characterized by a unitary representation $\rho_f(\gamma)$ of $\Gamma_N = \Gamma/\Gamma(N)$, and a weight $k_f$:

$$f_i(\gamma\tau) = (c\tau + d)^{k_f}\rho_f(\gamma)_{ij}f_j(\tau) \quad . \tag{4}$$

The integer $N$ is called the level of $f_i(\tau)$. For each level and for each even non-negative weight, there is only a finite number of linearly independent modular forms. For instance, for level 3 and weight 2, there are three linearly independent modular forms, which we will denote by $Y_i(\tau)$, transforming in the 3 representation of $\Gamma_3$ [20]. They are given in Appendix A.

By using these rules the superpotential of our models can be easily constructed and reads:

$$w = w_h + w_e + w_\nu \quad , \tag{5}$$

where $w_h$, $w_e$, $w_\nu$ describe the Higgs sector, the charged lepton sector and the neutrino sector, respectively. For both models the superpotential $w_e$ for the charged lepton sector reads:

$$w_e = -a\, E_1^c H_d(L\, \varphi)_1 - b\, E_2^c H_d(L\, \varphi)_{1'} - c\, E_3^c H_d(L\, \varphi)_{1''} \equiv -E^{cT}\mathcal{Y}_e H_d L \quad , \tag{6}$$

where $(...)_r$ denotes the $r$ representation of $\Gamma_3$. In the last equality we use a vector notation and

$$\mathcal{Y}_e = \begin{pmatrix} a\, \varphi_1 & a\, \varphi_3 & a\, \varphi_2 \\ b\, \varphi_2 & b\, \varphi_1 & b\, \varphi_3 \\ c\, \varphi_3 & c\, \varphi_2 & c\, \varphi_1 \end{pmatrix} \quad . \tag{7}$$

As a consequence of supersymmetry and modular invariance, this superpotential is completely determined. The dependence on the flavon supermultiplet $\varphi$ is linear, as required by the weight assignment. There is no dependence on the modulus $\tau$, since $(E_i^c H_d L)$ has weight $+3$ while the only combinations of $\tau$ with the required transformation properties under $\Gamma_3$ are modular forms with even non-vanishing weights. For both models the charged lepton mass matrix $m_e$ reads

$$m_e = \mathcal{Y}_e \frac{v}{\sqrt{2}}\cos\beta \quad . \tag{8}$$

In the neutrino sector $w_\nu$ depends on the model. In Model 1 neutrino masses originate from the Weinberg operator for which we have a unique possibility:

$$w_\nu = -\frac{1}{\Lambda}(H_u H_u\, LL\, Y)_1 \quad , \tag{9}$$

where $\Lambda$ stands for the scale associated to lepton number violation. In this case no dependence on the flavon $\varphi$ is allowed, whereas the invariance under the modular group requires $Y(\tau)$ to be the modular forms of level 3 and weight 2, collected in the Appendix A. Also $w_\nu$ is completely determined by supersymmetry and modular invariance. Apart for an overall scale, $w_\nu$ does not depend on any Lagrangian parameter.

In Model 2 light neutrinos get their masses from the see-saw mechanism and the terms of $w_\nu$ bilinear in the matter multiplets $L$ and $N^c$ read

$$w_\nu = -y_0(N^c H_u L)_1 + \Lambda(N^c N^c Y)_1 + ... \tag{10}$$

Dots denote terms containing three or more powers of the matter fields, such as for instance $(N^c)^3\varphi$, which have no impact on our analysis. At energies below the mass scale $\Lambda$ for both models we have:

$$w_\nu = -\frac{1}{\Lambda}(H_u L)^T \mathcal{W}(H_u L) + ... \quad , \tag{11}$$

with

$$\mathcal{W} = \begin{cases} \mathcal{C} & \text{Model 1} \\ \frac{1}{2}\left(\mathcal{Y}_\nu^T \mathcal{C}^{-1}\mathcal{Y}_\nu\right) & \text{Model 2} \end{cases} \quad . \tag{12}$$

Here $\mathcal{Y}_\nu$ stand for the constant matrix[5]

$$
\mathcal{Y}_\nu = y_0 \begin{pmatrix} 1 & 0 & 0 \\ 0 & 0 & 1 \\ 0 & 1 & 0 \end{pmatrix} \quad , \tag{13}
$$

whereas $\mathcal{C}$ denotes a matrix in generation space depending on the three independent level 3 and weight $+2$ modular forms $Y_i(\tau)$ ($i = 1, ..., 3$):

$$
\mathcal{C} = \begin{pmatrix} 2Y_1(\tau) & -Y_3(\tau) & -Y_2(\tau) \\ -Y_3(\tau) & 2Y_2(\tau) & -Y_1(\tau) \\ -Y_2(\tau) & -Y_1(\tau) & 2Y_3(\tau) \end{pmatrix} \quad . \tag{14}
$$

The light neutrino mass matrix $m_\nu$ is

$$
m_\nu = \mathcal{W} \frac{v^2}{\Lambda} \sin^2 \beta \quad . \tag{15}
$$

The parameters of the superpotential $w$ controlling lepton masses and mixing angles are the overall scale $\Lambda / \sin^2 \beta$ and the three dimensionless constants $a$, $b$ and $c$. They can all be made real, without loss of generality, by a phase transformation of the matter supermultiplets. As it is already evident from the structure of $w_e$, the three parameters $a$, $b$ and $c$ are in a one-to-one correspondence with the charged lepton masses. Electron, muon and tau masses cannot be predicted by the models, but just accommodated, exactly as in the Standard Model. All the remaining physical quantities, neutrino masses and lepton mixing angles, depend on a single Lagrangian parameter, the overall scale $\Lambda / \sin^2 \beta$, and on the vacuum structure of the theory.

Modular invariance of the kinetic terms can be easily achieved. A minimal choice for the Kähler potential $K(\Phi, \bar{\Phi})$ of Model 1 is

$$
\begin{aligned}
K(\Phi, \bar{\Phi}) = \quad & - \quad h \Lambda_K^2 \log(-i\tau + i\bar{\tau}) + \Lambda_K^2 (-i\tau + i\bar{\tau})^{-3} |\varphi|^2 + \sum_i (-i\tau + i\bar{\tau})^2 |E_i^c|^2 \\
& + \quad (-i\tau + i\bar{\tau}) |L|^2 + |H_u|^2 + |H_d|^2 \quad ,
\end{aligned} \tag{16}
$$

where $h$ is a positive constant, $\Lambda_K$ is a mass parameter. For Model 2 we take

$$
\begin{aligned}
K(\Phi, \bar{\Phi}) = \quad & - \quad h \Lambda_K^2 \log(-i\tau + i\bar{\tau}) + \Lambda_K^2 (-i\tau + i\bar{\tau})^{-3} |\varphi|^2 + \sum_i (-i\tau + i\bar{\tau})^4 |E_i^c|^2 \\
& + \quad (-i\tau + i\bar{\tau}) |N^c|^2 + (-i\tau + i\bar{\tau})^{-1} |L|^2 + |H_u|^2 + |H_d|^2 \quad .
\end{aligned} \tag{17}
$$

With such minimal choices, the transformations needed to put kinetic terms of matter super-fields in the canonical form can all be absorbed in a redefinition of the superpotential parameters and have no physical implications. Non-minimal Kähler potentials are allowed by the formalism, but they would in general introduce new input parameters affecting our predictions and reducing predictability. There are cases where non-minimal Kähler potentials have no consequences on our predictions. This occurs, for example, if the VEV of $\varphi$ and $Y(\tau)$ are sufficiently small. All our predictions do not depend on the absolute overall scale of such VEVs. In any case we consider the choice of minimal Kähler potential as part of the definition of our framework.

---

[5]At low energies the overall factor $y_0$ can be absorbed by a redefinition of $\Lambda$.

## 3 Fit to Leptonic Data

As shown in the previous section, all dimensionless quantities in the neutrino sector depend uniquely on the vacuum structure of the theory. To date there is no evidence for a specific mechanism choosing the vacuum of our world within the landscape of a hypothetical fundamental theory. Several key quantities related to the vacuum choice, such as the cosmological constant or the electroweak scale, have not yet found a convincing explanation. Thus we proceed by treating the modulus $\tau$ and the flavon VEVs as independent parameters, to be freely varied in order to adjust the agreement with the data. In this way we give up a possible dynamical justification of the adopted vacuum and we rely on some unspecified vacuum selection mechanism.

Having completely specified the models in the previous section, here we could simply quote the results of our fit to the experimental data, which we collect for convenience in table 3. However we find more instructive to illustrate the idea that has guided us in searching for the most promising region of the parameter space. We start by using the results of ref. [20] where the same models considered here were analyzed in a particular limit. Such a limit refers to the case where the charged lepton sector does not contribute to the lepton mixing matrix and corresponds to the special choice $\varphi = (1, 0, 0)$ of the present discussion. These results are shown in appendix B, from which we see that this limiting case can be considered as a fair first order approximation. Indeed, by a convenient choice of the complex modulus $\tau$, it leads to a reasonable agreement between theory and data as far as the neutrino squared mass differences and the mixing angles $\theta_{12}$ and $\theta_{23}$ are concerned. When $\varphi = (1, 0, 0)$ only the reactor angle deviates significantly from its experimental range. Even though in terms of standard deviations the disagreement is sharp, $\theta_{13}$ is predicted to be the smallest angle, between $12^0$ and $13^0$, not very far from the measured value. This partial result suggests that a better agreement with data can be obtained by exploring values of the flavon $\varphi$ close, but not identical, to $(1, 0, 0)$. We stress that this possibility is fully allowed by the considered models. As apparent from eq. (7), these values give rise to a non-diagonal mass matrix for the charged leptons, which can contribute to the lepton mixing and improve the first approximation. Thus here we perturb the pattern $\varphi = (1, 0, 0)$, by replacing the vanishing entries with small quantities. Aiming to a minimal amount of free parameters, here we turn on the real part of $\varphi_3$, which we simply denote by $\varphi_3$. This choice is motivated by the fact that in both models the reactor angle determined by assuming a diagonal charged lepton mass matrix remains essentially unchanged when turning on small imaginary parts of $\varphi_{2,3}$ and is mostly sensitive to the real part of $\varphi_3$, as shown in detail in Appendix B. Since the overall scale of $\varphi$ is a redundant parameter, without loss of generality we can set the flavon VEV to be $\varphi = (1, 0, \varphi_3)$. The induced non-diagonal charged lepton mass matrix contributes to the lepton mixing with the unitary transformation

$$U_e = \begin{pmatrix} 1 & \varphi_3 & 0 \\ 0 & -\varphi_3 & 1 \\ -\varphi_3 & 1 & \varphi_3 \end{pmatrix} + ... \tag{18}$$

where dots stand for terms of order $\varphi_3^2$, $(m_e^2/m_\mu^2)\varphi_3$ and $(m_\mu^2/m_\tau^2)\varphi_3$. It results from the combination of a small rotation and a permutation in the $(2, 3)$ sector.

The correct identification of the mass and mixing parameters requires also a transformation to put kinetic terms in the canonical form. It is easily seen from eqs. (16,17) that such transformations do not introduce any additional parameters beyond those already mentioned. In summary, our models have six dimensionless parameters: $a$, $b$, $c$, the complex modulus $\tau$ and $\varphi_3$. By varying them we would like to describe eleven physical quantities: the charged lepton Yukawa couplings, the neutrino mass ratios, the lepton mixing angles and the Dirac and Majorana phases. In this section we show the results of a scan of the parameter space where,

for each model, we look for the minimum of a $\chi^2$-function built with the data listed in table 3. In this first step of the analysis, the predictions of the two models do not include any further correction, such as running and supersymmetry breaking effects. The latter will be discussed separately in Sections 4 and 5.

Table 3: Left panel: charged lepton Yukawa couplings renormalized at the $m_Z$ scale, from ref. [52]. Right panel: neutrino oscillation data, from ref. [2]. The squared mass differences are defined as $\Delta m^2_{sol} = m^2_2 - m^2_1$ and $|\Delta m^2_{atm}| = |m^2_3 - (m^2_1 + m^2_2)/2|$. Errors are shown in brackets.

| $y_e(m_Z)$ | $2.794745(16) \times 10^{-6}$ |
|---|---|
| $y_\mu(m_Z)$ | $5.899863(19) \times 10^{-4}$ |
| $y_\tau(m_Z)$ | $1.002950(91) \times 10^{-2}$ |

| | IO | NO |
|---|---|---|
| $r \equiv \|\Delta m^2_{sol}/\Delta m^2_{atm}\|$ | 0.0301(8) | 0.0299(8) |
| $\sin^2 \theta_{12}$ | 0.303(13) | 0.304(13) |
| $\sin^2 \theta_{13}$ | 0.0218(8) | 0.0214(8) |
| $\sin^2 \theta_{23}$ | 0.56(3) | 0.55(3) |
| $\delta/\pi$ | 1.52(14) | 1.32(19) |

The results for Model 1 are displayed in table 4. The fit is excellent. All neutrino dimensionless observables fall in the $1\sigma$ experimental range. The minimum $\chi^2$, devided by two, the number of degrees of freedom, is less than 0.2. In parenthesis we show our error estimate. The errors do not correspond exactly to one standard deviation. For the input parameters, they have been computed by individually varying $\mathrm{Re}(\tau)$, $\mathrm{Im}(\tau)$ and $\varphi_3$ until the $\chi^2_{min}$ increases by one unit. For the observables, they have been computed by approximating the ellipsoid $\chi^2 = \chi^2_{min} + 1$ by the corresponding parallelepiped. For each observable, the error is defined as the difference between the maximum and the minimum values attained at the vertices, divided by two. The mass ordering is inverted. The atmospheric angle lies in the second octant. The Dirac phase is very close to $3\pi/2$. The model predicts three quantities that have not yet been measured: the ratio $m_3/m_2$, and the Majorana phases $\alpha_{21}$ and $\alpha_{31}$.

Also the absolute neutrino masses are determined, since their overall scale can be found by requiring that the individual square mass differences are reproduced. We find:

$$m_1 = 4.90(3) \times 10^{-2} \mathrm{eV} \quad , \qquad m_2 = 4.98(2) \times 10^{-2} \mathrm{eV} \quad , \qquad m_3 = 7.5(3) \times 10^{-4} \mathrm{eV} \quad , \quad (19)$$

where errors are displayed in parenthesis. By using these values and the predicted Majorana phases, we get

$$|m_{ee}| = 4.73(4) \times 10^{-2} \mathrm{eV} \quad , \tag{20}$$

for the mass combination entering neutrino-less double-beta decay. The parameter $\tau$ is close to the self-dual point $\tau = i$, the point where invariance under the $S$ transformation $\tau \to -1/\tau$ is preserved. The VEV of $\varphi_3$ is of order 0.1 and the charged lepton mass matrix is nearly diagonal, up to a permutation of the second and third generations. The parameters $a$, $b$, $c$ are tuned to reproduce the charged lepton Yukawa couplings.

The results for Model 2 are shown in table 5. The mass ordering is normal. There is a reasonable agreement with the data. The minimum $\chi^2$ is close to 10, dominated by the pull of $\sin^2 \theta_{23}$. Indeed the atmospheric angle is the largest one and close to maximal, but it lies in the first octant. Such a relatively high value of the $\chi^2$ is probably overestimated. In our fit we are treating all the errors as Gaussian and the central value of $\sin^2 \theta_{23}$ at the best fit

Table 4: Fit to data in Model 1, no RGE effects included. The neutrino mass spectrum has Inverted Ordering. Parameter values at the minimum of the $\chi^2$ and errors (in brackets) on the left panel. Best fit values, errors (in brackets) and pulls on the right panel. Pulls are defined as (best value−experimental value)/1$\sigma$. $\chi^2_{min} = 0.4$. For the definition of errors, see the text.

| | | | best value | pull |
|---|---|---|---|---|
| | | $r \equiv |\Delta m^2_{sol}/\Delta m^2_{atm}|$ | 0.0302(11) | +0.13 |
| | | $m_3/m_2$ | 0.0150(5) | − |
| | | $\sin^2 \theta_{12}$ | 0.304(17) | +0.08 |
| $\tau$ | $0.0117(4) + i\,0.9948(4)$ | $\sin^2 \theta_{13}$ | 0.0217(8) | −0.13 |
| $\varphi_3$ | $-0.086(4)$ | $\sin^2 \theta_{23}$ | 0.577(4) | +0.67 |
| $a\cos\beta$ | $2.806923 \times 10^{-6}$ | $\delta/\pi$ | 1.529(3) | +0.07 |
| $b\cos\beta$ | $9.992488 \times 10^{-3}$ | $\alpha_{21}/\pi$ | 0.135(6) | − |
| $c\cos\beta$ | $5.899778 \times 10^{-4}$ | $\alpha_{31}/\pi$ | 1.728(18) | − |
| | | $y_e(m_Z)$ | $2.794745 \times 10^{-6}$ | 0.0 |
| | | $y_\mu(m_Z)$ | $5.899864 \times 10^{-4}$ | +0.05 |
| | | $y_\tau(m_Z)$ | $1.002950 \times 10^{-2}$ | 0.0 |

point, $\sin^2 \theta_{23} = 0.46$, has a nominal deviation of $3\sigma$. In reality the error on $\sin^2 \theta_{23}$ is not Gaussian, as can be seen for instance in fig. 8 of ref. [2], top-left panel. The error on $\sin^2 \theta_{23}$ is asymmetrical and $\sin^2 \theta_{23} = 0.46$ is still within the $1\sigma$ range. Our $\chi^2$ minimisation does not account for this important feature. The fit is better than what indicated by the nominal minimum $\chi^2$ and probably as good as the one in Model 1. As before the ratio $m_3/m_2$, and the Majorana phases $\alpha_{21}$ and $\alpha_{31}$ are predictions of the model. Fitting also the overall scale, for neutrino masses and $|m_{ee}|$ we find:

$$m_1 = 1.09(3) \times 10^{-2} \text{eV} \quad , \qquad m_2 = 1.39(2) \times 10^{-2} \text{eV} \quad , \qquad m_3 = 5.11(4) \times 10^{-2} \text{eV} \quad , \quad (21)$$

$$|m_{ee}| = 1.04(2) \times 10^{-2} \text{eV} \quad . \tag{22}$$

Quite interestingly, the lightest neutrino has a mass close to 0.01 eV, resulting in a relatively large $|m_{ee}|$ for a normally ordered mass spectrum. The VEV of $\varphi_3$ is of order 0.1 also in this case. Once again the parameters $a$, $b$, $c$ are tuned to reproduce the charged lepton Yukawa couplings.

The results of our fits are completely dominated by $r$, $\sin^2 \theta_{12}$ and $\sin^2 \theta_{13}$. This can be verified by using as inputs only these data. In this case $\sin^2 \theta_{23}$ and $\delta_{CP}$ become predictions of the model. We have collected the corresponding outcome in Appendix C, for both models. All physical quantities do not appreciably vary by moving from the enlarged to the restricted set of input data.

Table 5: Fit to data in Model 2, no RGE effect included. The neutrino mass spectrum has Normal Ordering. Parameter values at the minimum of the $\chi^2$ and errors (in brackets) on the left panel. Best fit values, errors (in brackets) and pulls on the right panel. Pulls are defined as (best value−experimental value)/$1\sigma$. $\chi^2_{min} = 9.9$. For the definition of errors, see the text.

| | |
|---|---|
| $\tau$ | $-0.2005(18) + i\ 1.0578(56)$ |
| $\varphi_3$ | $0.117(4)$ |
| $a\cos\beta$ | $2.809569 \times 10^{-6}$ |
| $b\cos\beta$ | $9.961316 \times 10^{-3}$ |
| $c\cos\beta$ | $5.899455 \times 10^{-4}$ |

| | best value | pull |
|---|---|---|
| $r \equiv \lvert \Delta m^2_{sol}/\Delta m^2_{atm}\rvert$ | $0.0299(12)$ | $0.0$ |
| $m_3/m_2$ | $3.68(5)$ | $-$ |
| $\sin^2\theta_{12}$ | $0.306(11)$ | $+0.15$ |
| $\sin^2\theta_{13}$ | $0.0211(12)$ | $-0.42$ |
| $\sin^2\theta_{23}$ | $0.459(5)$ | $-3.04$ |
| $\delta/\pi$ | $1.438(8)$ | $+0.62$ |
| $\alpha_{21}/\pi$ | $1.704(5)$ | $-$ |
| $\alpha_{31}/\pi$ | $1.201(16)$ | $-$ |
| $y_e(m_Z)$ | $2.794745 \times 10^{-6}$ | $0.0$ |
| $y_\mu(m_Z)$ | $5.899863 \times 10^{-4}$ | $0.0$ |
| $y_\tau(m_Z)$ | $1.002950 \times 10^{-2}$ | $0.0$ |

We conclude this Section with a comment on the quark sector. From the above results, it is quite apparent that a unique modulus $\tau$ is inadequate to describe both the lepton and the quark sectors. Indeed, already the charged lepton sector seems to prefer a description in terms of a conventional flavon, rather then in terms of the same modulus describing neutrino masses. This can be understood looking at the dependence on $\tau$ of the modular forms involved in the present construction. Such a dependence is approximately exponential, when the imaginary part of $\tau$ is large. The hierarchy between the neutrino mass squared differences is mild and requires a modulus $\tau$ with an imaginary part close to one. To reproduce the large hierarchy observed in the quark sector or in the charged lepton sector, one or more moduli with larger imaginary parts are probably needed. To explore a more realistic model for both quark and lepton masses and mixing angles, we should conceivably introduce several independent moduli and analyze the modular properties of holomorphic functions of several variables. Such a possibility is also suggested by the low energy description of string compactifications.

## 4  Corrections from Supersymmetry Breaking

So far we have discussed our model in the unrealistic limit of exact supersymmetry. Supersymmetry can be acceptable only as a broken symmetry and our results should include corrections from supersymmetry breaking terms. There is no established theory of supersymmetry break-

ing. Even focussing on the case of supersymmetry broken by soft terms, the most general framework introduces plenty of new parameters. Our predictions can be trusted only if the corrections induced by the breaking terms are negligibly small. In this section we will show that a sizeable region of the parameter space where this is the case exists.

We assume that supersymmetry breaking can be parametrised by the $F$-component of a chiral superfield $X$, singlet under gauge and $\Gamma_3$ transformations and having a vanishing weight with respect to the modular group:

$$X = \theta^2 F \quad . \tag{23}$$

We denote by $M$ the characteristic scale with which the supersymmetry breaking sector communicates with the visible sector. By treating $X$ as a spurion and by using dimensional analysis, we can write down the supersymmetry breaking terms of our model. Soft supersymmetry breaking masses are generated at lowest order in the $1/M$ expansion and, for the supersymmetry breaking scale $m_{SUSY}$, we expect

$$m_{SUSY} \approx \frac{F}{M} \quad . \tag{24}$$

At order $1/M$ we have no corrections to either Yukawa couplings or to the Weinberg operator (or to right-handed neutrino masses in the seesaw case), because dimensions do not match. We regard $\Lambda$ as a spurion chiral superfield carrying two units of the total lepton number and we include suitable powers of $\Lambda$ to ensure vanishing total lepton number. All terms contributing to Yukawa interactions or to light neutrino masses (right-handed neutrino mass terms in the seesaw case) have dimension three, as a consequence of lepton number conservation and gauge invariance[6]. Therefore, at lowest order in $1/M$, corrections to such terms are of the type:

$$\delta S = \frac{1}{M^2} \int d^4x\, d^2\theta\, d^2\bar{\theta}\ X^\dagger\, f(\Phi, \bar{\Phi}) + h.c. \tag{25}$$

The function $f(\Phi, \bar{\Phi})$ is gauge and modular invariant and has mass dimension three. It is bilinear in the lepton superfields, it includes appropriate powers of the Higgs and $\Lambda$ superfields and can contain an arbitrary number of $\varphi$ and $\tau$ superfields. We will give the explicit form of $f(\Phi, \bar{\Phi})$ below, but already from eq. (25) we see that lepton masses get corrected by terms of relative order $F/M^2 \approx m_{SUSY}/M$. A sufficiently large hierarchy between $m_{SUSY}$ and $M$ can deplete such corrections to a negligible level. For example, if $M \approx 10^{18}$ GeV, even by taking $m_{SUSY}$ as large as $10^8$ GeV we get a correction to $\mathcal{Y}$ and $\mathcal{W}$ of order $10^{-10}$. Our predictions of neutrino mass ratios, mixing angles and phases are totally unaffected.

Focussing on Model 1, a first example of a contribution of the type (25) arises from holomorphic combination of the chiral supermultiplets $\Phi$ of the theory. Consider an holomorphic function $\Delta w(\Phi)$ of vanishing weight, invariant under gauge and $\Gamma_3$ transformations. Since we are only interested in lepton bilinear terms, $\Delta w(\Phi)$ should necessarily be of the same form as the superpotential $w(\Phi)$, eqs. (6-9), with new parameters $a'$, $b'$, $c'$. By choosing $f(\Phi, \bar{\Phi}) = \Delta w(\Phi)$, from eq. (25) we have

$$\begin{aligned} \delta S &= \frac{1}{M^2} \int d^4x\, d^2\theta\, d^2\bar{\theta}\ X^\dagger\ \Delta w(\Phi) + h.c. \\ &= \frac{F}{M^2} \int d^4x\, d^2\theta\ \Delta w(\Phi) + h.c. \end{aligned} \tag{26}$$

---

[6]Yukawa interactions require two lepton and a Higgs superfield. Interactions contributing to neutrino masses require two lepton plus two Higgs superfields and a $1/\Lambda$ insertion. Right-handed neutrino mass terms are bilinear in the lepton superfields and proportional to $\Lambda$.

Such a correction has no observable consequences, since it can be completely absorbed by redefining the parameters of the superpotential $w(\Phi)$. This example is a particular case of a more general choice:

$$f(\Phi, \bar{\Phi}) = (-i\tau + i\bar{\tau})^{-k} \, w_1(\Phi)\overline{w_2(\Phi)} \quad . \tag{27}$$

Here $w_{1,2}(\Phi)$ are holomorphic functions of equal weight $k$. The combination $w_1(\Phi)\overline{w_2(\Phi)}$ should be gauge and $\Gamma_3$ invariant, while this is not required for the individual functions $w_{1,2}(\Phi)$. Since we are looking for corrections to fermion masses, the function $w_1(\Phi)$ contains the bilinears $E_i^c L$ or $LL$, while the function $w_2(\Phi)$ depends only on $H_{u,d}$, $\varphi$ and $\tau$. The function in eq. (27) involving the minimum number of fields[7] is:

$$
\begin{aligned}
f(\Phi, \bar{\Phi}) = \quad & f_1 \, (E_1^c L \; \varphi)_1 H_u^\dagger + f_2 \, (E_2^c L \; \varphi)_1 H_u^\dagger + f_3 \, (E_3^c L \; \varphi)_1 H_u^\dagger + \\
& \frac{f_4}{\Lambda}(LL \; Y)_1 H_d^\dagger H_d^\dagger + \frac{f_5}{\Lambda}(LL \; YH_u)_1 H_d^\dagger \quad .
\end{aligned}
\tag{28}
$$

After integration over the Grassmann variables, we get:

$$
\begin{aligned}
\delta S = \frac{F}{M^2} \int d^4 x \; \Big[ \quad & f_1 \, (e_1^c \ell \; \varphi)_1 H_u^\dagger + f_2 \, (e_2^c \ell \; \varphi)_1 H_u^\dagger + f_3 \, (e_3^c l \; \varphi)_1 H_u^\dagger + \\
& \frac{f_4}{\Lambda}(\ell\ell \; Y)_1 H_d^\dagger H_d^\dagger + \frac{f_5}{\Lambda}(\ell\ell \; YH_u)_1 H_d^\dagger \Big] \quad ,
\end{aligned}
\tag{29}
$$

where here $H_{u,d}$, $\varphi$ and $Y$ denote the physical scalar components of the chiral multiplets, whereas $e_k^c$ and $\ell$ stand for the corresponding fermions. Also this terms gives rise to corrections that, as far as fermion masses are concerned, can be absorbed in a redefinition of the input parameters. Non-minimal choices of $f(\Phi, \bar{\Phi})$ give rise to corrections that cannot be incorporated in the initial input parameters, but their contribution to the $\mathcal{Y}$ and $\mathcal{W}$ couplings is of order $m_{SUSY}/M$. A similar discussion applies to Model 2. We conclude that these corrections can be safely neglected provided the separation between the two scales $m_{SUSY}$ and $M$ is sufficiently large. This requirement allows a wide possible range for the effective SUSY breaking scale $m_{SUSY}$, in particular not excluding the possibility of $m_{SUSY}$ much higher than the TeV scale.

Supersymmetry breaking can also affect our predictions through threshold corrections to lepton masses and mixing angles, when the theories above and below the $m_{SUSY}$ scale are matched. We comment such effect in the next section.

## 5 Corrections from Renormalization Flow

An important source of corrections is represented by radiative corrections, which can be enhanced by the gap between the scale $\Lambda$, here assumed to be very large, and the electroweak scale, which we identify with the $Z$ mass $m_Z$. The large logarithms arising in these corrections can be efficiently resummed through well-known renormalization group (RGE) techniques, which we summarize in Appendix D. Here we describe the main steps of the analysis and we collect our final results. RGE corrections are expected to be relevant when a degeneracy between neutrino masses occur, which in our case mainly applies to the inverted ordering case of Model 1.

---

[7]The most general correction involve combinations $w_1(\Phi)\overline{w_2(\Phi)}$ such as $[E^c L \; \varphi^p Y^q (\varphi^\dagger)^{p-1+2r} (Y^\dagger)^{q-3r}]$ and $[LL \; \varphi^m Y^n (\varphi^\dagger)^{m-2s} (Y^\dagger)^{n-1+3s}]$, where $p$, $q$, $m$, $n$ are non-negative integers, $r$ and $s$ are relative integers and the combinations $p-1+2r$, $q-3r$, $m-2s$, $n-1+3s$ are non-negative. The functions $w_{1,2}(\Phi)$ have weights equal to $(3p+2q-3)$ and $(3m+2n-2)$, respectively. The dependence of $H_{u,d}$, which have vanishing weight, can be easily included. The terms in eq. (29) refer to the choice $r = s = q = m = 0$, $p = n = 1$.

In Model 1 charged lepton Yukawa couplings $\mathcal{Y}_e$, eq. (6), and the matrix $\mathcal{W}$ characterising the Weinberg operator, eq. (12), are predicted at the scale $\Lambda$ as functions of our input parameters $a, b, c, \tau, \varphi_3$. It is convenient to work in the basis where the charge lepton Yukawa couplings are diagonal, since this basis is preserved by the running. Quantities in this basis will be denoted by a hat. Thus $\hat{\mathcal{Y}}_e(\Lambda)$ and $\hat{\mathcal{W}}(\Lambda)$ provide the boundary conditions for the RGE flow. By solving the RGE equations in the unbroken supersymmetric case we get $(\hat{\mathcal{Y}}_e)_{MSSM}(m_{SUSY})$ and $(\hat{\mathcal{W}})_{MSSM}(m_{SUSY})$. They depend on gauge and Yukawa couplings, on $\Lambda$ and $\tan\beta$. By requiring the continuity of the physical quantities, i.e. the charged lepton masses and the neutrino masses, we get the analogous quantities referred to the SM, where all supersymmetric particles have been integrated out. At lowest order, that is by neglecting threshold corrections [8] at the scale $m_{SUSY}$, the matching conditions are given by:

$$
\begin{aligned}
(\hat{\mathcal{Y}}_e)_{SM}(m_{SUSY}) &= (\hat{\mathcal{Y}}_e)_{MSSM}(m_{SUSY})\cos\beta \quad, \\
(\hat{\mathcal{W}})_{SM}(m_{SUSY}) &= (\hat{\mathcal{W}})_{MSSM}(m_{SUSY})\sin^2\beta \quad.
\end{aligned}
\tag{30}
$$

The final values of $\hat{\mathcal{Y}}_e(m_Z)$ and $\hat{\mathcal{W}}(m_Z)$ at the scale $m_Z$ depend on the input parameters and on $\Lambda$, $m_{SUSY}$ and $\tan\beta$. We work in the one-loop approximation for beta functions and anomalous dimensions.

Table 6: Minimum $\chi^2$ and best fit values of $(r, \sin^2\theta_{12})$ as a function of $\tan\beta$ and $m_{SUSY}$ for Model 1. The mass ordering is inverted. We set $\Lambda = 10^{15}$ GeV. Full results are collected in Appendix E.

| $m_{SUSY}$ | Quantity | $\tan\beta = 2.5$ | $\tan\beta = 10$ | $\tan\beta = 15$ |
|---|---|---|---|---|
| $10^4$ GeV | $r$ | 0.0302 | 0.0292 | 0.0288 |
| | $\sin^2\theta_{12}$ | 0.304 | 0.345 | 0.418 |
| | $\chi^2_{min}$ | 0.4 | 12.2 | 82.0 |
| $10^8$ GeV | $r$ | 0.0302 | 0.0294 | 0.0286 |
| | $\sin^2\theta_{12}$ | 0.303 | 0.335 | 0.389 |
| | $\chi^2_{min}$ | 0.4 | 7.0 | 47.7 |

In Model 1 the RGE flows mainly affects $r$, through its dependence on $\Delta m^2_{21}$, and $\sin^2\theta_{12}$. In table 6 we show the results of a fit to our input parameters $a, b, c, \tau, \varphi_3$, for several values of $\tan\beta$ and $m_{SUSY}$. We set $\Lambda = 10^{15}$ GeV. We have reported only the best fit values of $(r, \sin^2\theta_{12})$ and the minimum $\chi^2$, while full results are given in Appendix E. Model 1 keeps providing a good fit for relatively small values of $\tan\beta$ and for high values of $m_{SUSY}$, but the agreement with data degrades for large $\tan\beta$, especially when $m_{SUSY}$ is relatively small. When $\tan\beta$ and $m_{SUSY}$ are varied, the best fit values of $(\tau, \varphi_3)$ remain relatively stable, while $r$ and $\sin^2\theta_{12}$ deviate from the experimentally allowed region. This can be qualitatively understood by inspecting the RGE equations for $r$ and $\sin^2\theta_{12}$ in the one-loop approximation [53]. Their approximate analytical expressions read:

$$
\begin{aligned}
\frac{dr}{dt} &= \frac{y_\tau^2}{4\pi^2} r \sin^2\theta_{23} \frac{m_1^2(\cos^2\theta_{12} - \sin^2\theta_{12})}{m_2^2 - m_1^2} + \ldots \\
\frac{d\sin^2\theta_{12}}{dt} &= -\frac{y_\tau^2}{32\pi^2}\sin^2 2\theta_{12}\sin^2\theta_{23}\frac{|m_1 + m_2 e^{-i\alpha_{21}}|^2}{m_2^2 - m_1^2} + \ldots
\end{aligned}
\tag{31}
$$

---

[8] We comment on threshold effects at the and of this section.

where dots stand for sub-leading contributions. From high to low energies $\sin^2\theta_{12}$ increases, while $r$ decreases as long as $\sin^2\theta_{12} < 0.5$. In fig. 1 we show contour lines of $\chi^2_{min}$ in the plane $(\tan\beta, m_{SUSY})$.

This establishes that a non-negligible region in parameter space exists where the agreement between prediction and data is excellent. Low values of $\tan\beta$ and, to less extent, large values of $m_{SUSY}$ are preferred. The fit to neutrino data is not good when $\tan\beta$ exceeds 10.

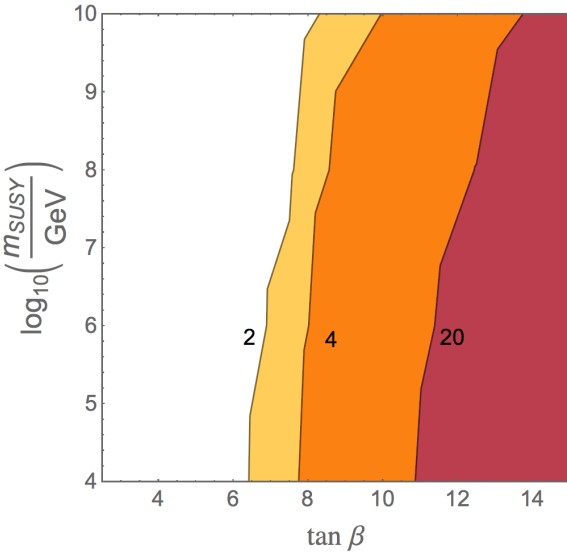

Figure 1: Contours of equal minimum $\chi^2$ in the plane $(\tan\beta, m_{SUSY})$.

In the above analysis we have enforced the lowest-order matching condition at the scale $m_{SUSY}$, eq. (30), neglecting threshold corrections. These depend on the details of the SUSY mass spectrum, which is not entirely under our control, given the ignorance on the SUSY breaking mechanism. To estimate threshold effects, we consider the simple case of degenerate left-handed sleptons and negligible mixing both between left and right sectors and in each individual sector. Such a spectrum is consistent with our flavour symmetry. Indeed, for sufficiently large $m_{SUSY}$ the mixing between left-handed and right-handed sleptons, roughly proportional to $(m_\ell \tan\beta/m_{SUSY})$ $(\ell = e, \mu, \tau)$, can be made very small. Moreover, degenerate left-handed sleptons originate by insertion of the spurion $X^\dagger X$ into the minimal form of the Kähler potential, eqs. (16) and (17). Nearly diagonal right-handed sleptons are expected if the $E_i^c$ superfields carry an additional charge to the purpose of explaining the hierarchy of the parameters $a$, $b$ and $c$. More general SUSY spectra are also allowed in our framework.

The matching equations between MSSM and SM charged lepton Yukawa couplings $\hat{\mathcal{Y}}_e$ are corrected by terms of order $(1 + \epsilon_\ell \tan\beta)$ $(\ell = e, \mu, \tau)$, with $\epsilon_\ell \approx 10^{-3}$ [54]. In our case the flavour-diagonal corrections $\epsilon_\ell$ are almost universal. The breaking of universality is sourced by the right-handed slepton sector, that typically gives a subdominant contribution to $\epsilon_\ell$ proportional to $g_Y^2$. Such corrections can be compensated by varying the input parameters $a$, $b$, $c$, without affecting the neutrino data. Similarly, the correction to the matching equation for the neutrino coupling $\hat{\mathcal{W}}$ depends mainly on left-handed slepton masses [55]. Even though such correction can be of relative order $10^{-2}$, for degenerate left-handed sleptons it is almost universal and mostly affects the overall scale of $\mathcal{W}$, with no appreciable modifications of our predictions.

In Model 2 the corrections from RGE are very small. As an example, in table 7 we show the comparison between the values of neutrino parameters obtained with and without including running effects. Data at $\Lambda = 10^{15}$ GeV refer to the best fit values of table 5 and do not account for the RGE flow. Data at the scale $m_Z$ have been obtained by solving the RGE in the one-

Table 7: RGE evolution for Model 2. Data at $\Lambda = 10^{15}$ GeV are from the best fit values of table 5. Data at the scale $m_Z$ are from the RGE flow for $\tan\beta = 25$ and $m_{SUSY} = 10^4$ GeV.

|  | $\Lambda = 10^{15}$ GeV | $m_Z$ |
|---|---|---|
| $r \equiv \lvert \Delta m^2_{sol}/\Delta m^2_{atm} \rvert$ | 0.0299 | 0.0299 |
| $m_3/m_2$ | 3.68 | 3.67 |
| $\sin^2\theta_{12}$ | 0.306 | 0.305 |
| $\sin^2\theta_{13}$ | 0.0211 | 0.0212 |
| $\sin^2\theta_{23}$ | 0.46 | 0.46 |
| $\delta/\pi$ | 1.44 | 1.44 |
| $\alpha_{21}/\pi$ | 1.70 | 1.70 |
| $\alpha_{31}/\pi$ | 1.20 | 1.19 |

loop approximation for $\tan\beta = 25$ and $m_{SUSY} = 10^4$ GeV. We have not included the running from the UV scale $\Lambda_{UV}$ and the right-handed mass threshold $M_1 \equiv \Lambda$ since, as explained in Appendix D, its effect amounts to an overall rescaling of $\mathcal{W}$, to a good approximation. We have performed a new optimization of our input parameters, by minimizing the new $\chi^2$-function that now incorporates the RGE effects. We see that the predictions are very stable. For smaller values of $\tan\beta$ and higher values of $m_{SUSY}$ we expect an even smaller effect. We conclude that the results for Model 2 are not appreciably modified by the running from the UV scale down to low energy, for a wide range of $(\tan\beta, m_{SUSY})$.

## 6 Conclusion

The very good experimental accuracy recently achieved in neutrino oscillation data calls for theoretical frameworks with an adequate level of predictability. As a step forward in the realization of this program, an important ingredient can be represented by modular invariance. Such a symmetry is so strong that in some cases the Yukawa interactions are completely determined in terms of a complex scalar field, the modulus. A concrete modular invariant framework for lepton masses has been recently proposed in ref. [20], with two specific realizations. In Model 1 neutrinos get their masses from the Weinberg operator, while in Model 2 neutrino masses come from the seesaw mechanism. Apart from charged lepton Yukawa couplings, that requires three ad-hoc parameters to be correctly reproduced, all the remaining dimensionless quantities of the lepton sector do not depend on any Lagrangian parameter, but only on the vacuum structure of the model. In these models the vacuum can be minimally parametrized by a complex modulus, the building block of modular transformations, and a real VEV describing the small departure of the charged lepton mass matrix from the diagonal pattern. Thus eight independent physical quantities, such as the neutrino mass ratios, the three mixing angles, the Dirac and Majorana phases, are completely determined in terms of three real parameters. While this framework presents a close conceptual similarity with the Froggatt-Nielsen proposal [68], we stress that in the models analyzed here there are no adjustable parameters for the neutrino mass matrix, apart from an overall constant. On the contrary in the Froggatt-Nielsen scenario there is an independent parameter for each entry of the mass matrix, thus preventing predictions beyond the order-of-magnitude level.

To date we have little clue on how to chose the vacuum of our world within the possible landscape of a fundamental theory. The cosmological constant and the electroweak scale are two examples of vacuum-related quantities that have not yet found a convincing explanation.

Therefore in our work we have reversed the paradigm that requires the flavon VEVs to be determined by the minimum of the energy density and we have simply scanned the possible vacua in search of a configuration that maximizes the agreement between data and theory. For both realizations we have been able to find a vacuum that leads to a very good fit to neutrino data. Model 1 predicts an inverted mass ordering and has the best $\chi^2_{min}$, while Model 2 has a normal mass ordering and a reasonable $\chi^2_{min}$.

Though we have not attempted to select the vacuum through some dynamical mechanism, it is very intriguing that, for both models, data prefer vacua supporting specific residual symmetries. The modulus $\tau$, relevant to the neutrino sector, is in the vicinity of the self-dual point $i$ where the $S$ transformation $\tau \to -1/\tau$ is preserved. The alignment of the flavon $\varphi$, which controls charged leptons, is not far from $(1, 0, 0)$, where the group generated by $T$ is unbroken.

Supersymmetry and its breaking lead to a possible dependence of our results on several additional parameters such as the supersymmetry breaking scale $m_{SUSY}$, the messenger scale $M$, the cutoff scale $\Lambda$ (or $\Lambda_{UV}$) and $\tan\beta$. We have carefully studied such dependence. We have shown that supersymmetry breaking contributions to Yukawa couplings and neutrino masses scale as $m_{SUSY}/M$ and thus they can be safely neglected if there is a large gap between $m_{SUSY}$ and $M$. This represents a mild condition, that can be satisfied for a very wide range of $m_{SUSY}$. Moreover, by analyzing the RGE dependence of our results, we found that for Model 2 this is essentially negligible in a wide portion of the $(m_{SUSY}, \tan\beta)$ plane, while for Model 1 the dependence is stronger and the agreement between theory and data is spoiled for large $\tan\beta$. Nevertheless there is a finite region of the plane $(m_{SUSY}, \tan\beta)$, roughly given by $\tan\beta < 10$, where Model 1 still provides a good description of the data.

We stress that, once the right vacuum has been chosen, neutrino masses and lepton mixing parameters are all determined. All measured parameters are well reproduced. Moreover the Majorana phases are predicted, the absolute value of neutrino masses can be determined by reproducing the individual squared mass differences and the mass combination relevant to neutrinoless double-beta decay is fixed. We consider these properties as significant hints in favour of modular invariance as a guiding principle towards the solution of the flavour puzzle.

# Acknowledgements

This project has received support in part by the MIUR-PRIN project 2015P5SBHT 003 "Search for the Fundamental Laws and Constituents" and by the European Union's Horizon 2020 research and innovation programme under the Marie Sklodowska-Curie grant agreement N° 674896 and 690575. The research of F. F. was supported in part by the INFN. The research of J. C. C. was supported by the Spanish MINECO project FPA2016-78220-C3-1-P, the Junta de Andalucía grant FQM101 and the Spanish MECD grant FPU14.

# Appendices

## A  Modular Forms of Level 3 and Weight 2

Modular forms of level 3 form a ring generated by three linearly independent forms of weight 2. These are given by [20]:

$$
\begin{aligned}
Y_1(\tau) &= \frac{i}{2\pi}\left[\frac{\eta'\left(\frac{\tau}{3}\right)}{\eta\left(\frac{\tau}{3}\right)} + \frac{\eta'\left(\frac{\tau+1}{3}\right)}{\eta\left(\frac{\tau+1}{3}\right)} + \frac{\eta'\left(\frac{\tau+2}{3}\right)}{\eta\left(\frac{\tau+2}{3}\right)} - \frac{27\eta'(3\tau)}{\eta(3\tau)}\right] \\
Y_2(\tau) &= \frac{-i}{\pi}\left[\frac{\eta'\left(\frac{\tau}{3}\right)}{\eta\left(\frac{\tau}{3}\right)} + \omega^2 \frac{\eta'\left(\frac{\tau+1}{3}\right)}{\eta\left(\frac{\tau+1}{3}\right)} + \omega \frac{\eta'\left(\frac{\tau+2}{3}\right)}{\eta\left(\frac{\tau+2}{3}\right)}\right] \\
Y_2(\tau) &= \frac{-i}{\pi}\left[\frac{\eta'\left(\frac{\tau}{3}\right)}{\eta\left(\frac{\tau}{3}\right)} + \omega \frac{\eta'\left(\frac{\tau+1}{3}\right)}{\eta\left(\frac{\tau+1}{3}\right)} + \omega^2 \frac{\eta'\left(\frac{\tau+2}{3}\right)}{\eta\left(\frac{\tau+2}{3}\right)}\right]\ ,
\end{aligned}
\tag{32}
$$

where $\eta(\tau)$ is the Dedekind eta-function, defined in the upper complex plane:

$$
\eta(\tau) = q^{1/24}\prod_{n=1}^{\infty}(1-q^n) \qquad q \equiv e^{i2\pi\tau}\ .
\tag{33}
$$

They transform in the three-dimensional representation of $\Gamma_3$. By defining $Y^T = (Y_1, Y_2, Y_3)$ we have

$$
Y(-1/\tau) = \tau^2\, \rho(S)Y(\tau)\ , \qquad Y(\tau+1) = \rho(T)Y(\tau)\ ,
$$

with unitary matrices $\rho(S)$ and $\rho(T)$

$$
\rho(S) = \frac{1}{3}\begin{pmatrix} -1 & 2 & 2 \\ 2 & -1 & 2 \\ 2 & 2 & -1 \end{pmatrix}\ , \qquad \rho(T) = \begin{pmatrix} 1 & 0 & 0 \\ 0 & \omega & 0 \\ 0 & 0 & \omega^2 \end{pmatrix}\ , \qquad \omega = -\frac{1}{2} + \frac{\sqrt{3}}{2}i\ .
$$

The $q$-expansion of $Y_i(\tau)$ is given by:

$$
\begin{aligned}
Y_1(\tau) &= 1 + 12q + 36q^2 + 12q^3 + \dots \\
Y_2(\tau) &= -6q^{1/3}(1 + 7q + 8q^2 + \dots) \\
Y_3(\tau) &= -18q^{2/3}(1 + 2q + 5q^2 + \dots)\ .
\end{aligned}
$$

Moreover $Y_i(\tau)$ satisfy the constraint:

$$
Y_2^2 + 2Y_1Y_3 = 0\ .
\tag{34}
$$

# B  Dependence on $\varphi = (1, \varphi_2, \varphi_3)$

A fit to the data assuming the pattern $\varphi = (1, 0, 0)$, was carried out in ref. [20]. Such a pattern preserves the subgroup generated by $T$ and produces a diagonal charged lepton mass matrix. In this case, for Model 1 the value of $\tau$ that maximizes the agreement with the data, see table 3, is $\tau = 0.0116 + 0.9946i$. This value is very close to the self-dual point $\tau = i$, where the $S$ generator of the modular group is unbroken. The choice $\tau = 0.0116 + 0.9946i$ gives rise to the results collected in table 8.

Table 8: Fit to data in Model 1, with $\varphi = (1, 0, 0)$ and no RGE effects included. The neutrino mass spectrum has Inverted Ordering. Pulls are defined as (best value−experimental value)/$1\sigma$.

| | best value | pull | | best value | pull |
|---|---|---|---|---|---|
| $r \equiv \|\Delta m^2_{sol}/\Delta m^2_{atm}\|$ | 0.0302 | +0.13 | $\sin^2 \theta_{23}$ | 0.65 | +3.0 |
| $m_3/m_2$ | 0.0150 | − | $\delta/\pi$ | 1.55 | +0.07 |
| $\sin^2 \theta_{12}$ | 0.302 | −0.08 | $\alpha_{21}/\pi$ | 0.22 | − |
| $\sin^2 \theta_{13}$ | 0.0447 | +28.6 | $\alpha_{31}/\pi$ | 1.79 | − |

We see that the major source of disagreement is represented by the value of $\sin^2 \theta_{13}$ and, to a less extent, that of $\sin^2 \theta_{23}$. If we keep $\tau$ fix and, at the same time, we allow for a small correction originating from the general VEV $\varphi = (1, \varphi_2, \varphi_3)$, $r$ does not change and, to first order in the complex parameters $\varphi_2$ and $\varphi_3$, we find:

$$
\begin{aligned}
\sin^2 \theta_{12} &= 0.302 + 0.55 \, \text{Im}(\varphi_2) + 0.75 \, \text{Im}(\varphi_3) + 0.08 \, \text{Re}(\varphi_2) + 0.11 \, \text{Re}(\varphi_3) + \dots \\
\sin^2 \theta_{13} &= 0.0447 + 0.33 \, \text{Re}(\varphi_2) - 0.24 \, \text{Re}(\varphi_3) + \dots \\
\sin^2 \theta_{23} &= 0.65 - 0.12 \, \text{Re}(\varphi_2) - 1.12 \, \text{Re}(\varphi_3) + \dots \\
\delta/\pi &= 1.55 + 1.2 \, \text{Im}(\varphi_2) - 1.5 \, \text{Im}(\varphi_3) + 0.2 \, \text{Re}(\varphi_2) + 0.2 \, \text{Re}(\varphi_3) + \dots \\
\alpha_{21}/\pi &= 0.22 - 0.1 \, \text{Im}(\varphi_2) - 0.2 \, \text{Im}(\varphi_3) + 0.8 \, \text{Re}(\varphi_2) + 1.1 \, \text{Re}(\varphi_3) + \dots \\
\alpha_{31}/\pi &= 1.79 - 0.1 \, \text{Im}(\varphi_2) - 1.3 \, \text{Im}(\varphi_3) + 0.6 \, \text{Re}(\varphi_2) + 0.8 \, \text{Re}(\varphi_3) + \dots
\end{aligned}
\tag{35}
$$

where dots stand either for very small terms or higher-order contributions. It is interesting to see that, to first order in $\varphi_{2,3}$, $\sin^2 \theta_{13}$ is corrected only by the real parts. Moreover $\text{Re}(\varphi_3)$ provides the good correlation to adjust also $\sin^2 \theta_{23}$. Similar considerations can be done also for Model 2: for $\varphi = (1, 0, 0)$ the best fit value of $\sin^2 \theta_{13}$ is too large. Moreover there is a very small dependence of $\theta_{13}$ on $\text{Im}(\varphi_{2,3})$, which is mostly sensitive to $\text{Re}(\varphi_3)$.

## C   Fit to Reduced Data Set

We collect here the results of a fit to Model 1 and Model 2, where we use as input data only $r$, $\sin^2\theta_{12}$ and $\sin^2\theta_{13}$.

Table 9: Fit to $r$, $\sin^2\theta_{12}$ and $\sin^2\theta_{13}$ in Model 1, no RGE effects included. The neutrino mass spectrum has Inverted Ordering. Parameter values at the minimum of the $\chi^2$ on the left panel. Best fit values and pulls on the right panel.

| | |
|---|---|
| $\tau$ | $0.0117 + i\,0.9948$ |
| $\varphi_3$ | $-0.085$ |
| $a\cos\beta$ | $2.806619 \times 10^{-6}$ |
| $b\cos\beta$ | $9.993340 \times 10^{-3}$ |
| $c\cos\beta$ | $5.899783 \times 10^{-4}$ |

| | best value | pull |
|---|---|---|
| $r \equiv \lvert \Delta m^2_{sol}/\Delta m^2_{atm} \rvert$ | 0.0302 | +0.13 |
| $m_3/m_2$ | 0.0150 | — |
| $\sin^2\theta_{12}$ | 0.304 | +0.08 |
| $\sin^2\theta_{13}$ | 0.0219 | +0.13 |
| $\sin^2\theta_{23}$ | 0.578 | +0.67 |
| $\delta/\pi$ | 1.529 | +0.07 |
| $\alpha_{21}/\pi$ | 0.136 | — |
| $\alpha_{31}/\pi$ | 1.729 | — |
| $y_e(m_Z)$ | $2.794745 \times 10^{-6}$ | 0.0 |
| $y_\mu(m_Z)$ | $5.899863 \times 10^{-4}$ | 0.0 |
| $y_\tau(m_Z)$ | $1.002950 \times 10^{-2}$ | 0.0 |

Table 10: Fit to $r$, $\sin^2\theta_{12}$ and $\sin^2\theta_{13}$ in Model 2, no RGE effect included. The neutrino mass spectrum has Normal Ordering. Parameter values at the minimum of the $\chi^2$ on the left panel. Best fit values and pulls on the right panel.

| | best value | pull |
|---|---|---|
| $r \equiv \lvert \Delta m^2_{sol}/\Delta m^2_{atm}\rvert$ | 0.0299 | 0.0 |
| $m_3/m_2$ | 3.68 | — |
| $\sin^2\theta_{12}$ | 0.304 | 0.0 |
| $\sin^2\theta_{13}$ | 0.0214 | 0.0 |
| $\sin^2\theta_{23}$ | 0.457 | −3.0 |
| $\delta/\pi$ | 1.437 | +0.62 |
| $\alpha_{21}/\pi$ | 1.706 | — |
| $\alpha_{31}/\pi$ | 1.199 | — |
| $y_e(m_Z)$ | $2.794745 \times 10^{-6}$ | 0.0 |
| $y_\mu(m_Z)$ | $5.899863 \times 10^{-4}$ | 0.0 |
| $y_\tau(m_Z)$ | $1.002950 \times 10^{-2}$ | 0.0 |

| | |
|---|---|
| $\tau$ | $-0.2003 + i\,1.0589$ |
| $\varphi_3$ | 0.115 |
| $a\cos\beta$ | $2.809135 \times 10^{-6}$ |
| $b\cos\beta$ | $9.963605 \times 10^{-3}$ |
| $c\cos\beta$ | $5.899487 \times 10^{-4}$ |

## D  RGE Equations

We collect here formulas relevant for the analysis of the scale dependence of lepton mass and mixing parameters.

**Neutrino masses from the Weinberg operator**

In this case the lepton sector is defined by the Lagrangian:

$$\mathcal{L}_{\text{lept}} = -E^c \mathcal{Y}_e H^\dagger L + \frac{\mathcal{W}}{\Lambda}(\tilde{H}^\dagger L)(\tilde{H}^\dagger L) + h.c. \quad , \tag{36}$$

with the replacement $H^\dagger \to H_d$ and $\tilde{H}^\dagger \to H_u$ going from SM to MSSM. The lepton mass matrices are given by

$$m_\ell = \mathcal{Y}_e \frac{v}{\sqrt{2}}(\cos\beta)^p \quad , \qquad m_\nu = -\frac{\mathcal{W}}{\Lambda}v^2(\sin^2\beta)^p \quad , \tag{37}$$

where $p = 0$ in the SM and $p = 1$ in the MSSM.

Working in the one-loop approximation, the RGE equations for neutrinos [56–60] and charged lepton mass parameters [61–65] read

$$16\pi^2\frac{d\mathcal{Y}_e}{dt} = \mathcal{Y}_e\left[\hat{a}\,\mathcal{Y}_e^\dagger\mathcal{Y}_e + \text{tr}(\mathcal{Y}_e^\dagger\mathcal{Y}_e) + u\right] \tag{38}$$

$$16\pi^2\frac{d\mathcal{W}}{dt} = k\,\mathcal{W} + \hat{b}\left(\mathcal{W}\,(\mathcal{Y}_e^\dagger\mathcal{Y}_e) + (\mathcal{Y}_e^\dagger\mathcal{Y}_e)^T\,\mathcal{W}\right) \quad , \tag{39}$$

where

$$\hat{a} = \begin{cases} +3/2 & \text{SM} \\ +3 & \text{MSSM} \end{cases} , \tag{40}$$

$$u = \begin{cases} -\frac{15}{4}g_Y^2 - \frac{9}{4}g_2^2 + 3\,\text{tr}(\mathcal{Y}_u^\dagger\mathcal{Y}_u + \mathcal{Y}_d^\dagger\mathcal{Y}_d) & \text{SM} \\ -3(g_Y^2 + g_2^2) + 3\,\text{tr}(\mathcal{Y}_d^\dagger\mathcal{Y}_d) & \text{MSSM} \end{cases} , \tag{41}$$

$$\hat{b} = \begin{cases} -3/2 & \text{SM} \\ +1 & \text{MSSM} \end{cases} , \tag{42}$$

$$k = \begin{cases} -3g_2^2 + 6\,\text{tr}(\mathcal{Y}_u^\dagger\mathcal{Y}_u + \mathcal{Y}_d^\dagger\mathcal{Y}_d + \frac{1}{3}\mathcal{Y}_e^\dagger\mathcal{Y}_e) + 4\lambda & \text{SM} \\ -2g_Y^2 - 6g_2^2 + 6\,\text{tr}(\mathcal{Y}_u^\dagger\mathcal{Y}_u) & \text{MSSM} \end{cases} . \tag{43}$$

Here $\lambda$ is the Higgs self-coupling as defined by the quartic term in the scalar potential $-\lambda(H^\dagger H)^2$. Note that here we are using $g_Y$, the SM gauge coupling, related to $g_1$, used in the GUT context, by $g_1^2 = 5/3g_Y^2$. It is not restrictive to use a basis, preserved by the RGE flow, where the Yukawa couplings $\mathcal{Y}_e$ are diagonal: $\mathcal{Y}_e = \text{diag}(y_e, y_\mu, y_\tau)$. We get simple analytical solutions of these equations within the following approximations:

- We neglect the $\mu$ dependence of the combinations $u$ and $k$ in the LHS of eq. (38,39).

- In the equations for $y_i$ ($i = e, \mu, \tau$) we neglect the terms proportional to $y_i y_{e,\mu}^2$ compared to those proportional to $y_i y_\tau^2$.

The analytical solutions read:

$$y_i(\mu) = y_i(\mu_0)\left(\frac{\mu}{\mu_0}\right)^{\frac{y_\tau^2(\mu_0) + u}{16\pi^2}} \qquad (i = e, \mu) \tag{44}$$

$$y_\tau^2(\mu) = y_\tau^2(\mu_0)\frac{1}{\frac{(\hat{a}+1)}{u}y_\tau^2(\mu_0)\left(\left(\frac{\mu_0}{\mu}\right)^{\frac{u}{8\pi^2}} - 1\right) + \left(\frac{\mu_0}{\mu}\right)^{\frac{u}{8\pi^2}}} \tag{45}$$

$$\mathcal{W}_{ij}(\mu) = \mathcal{W}_{ij}(\mu_0)I_K I_i I_j , \tag{46}$$

where we have defined

$$I_K = \exp\left[\int_0^t dt' \frac{k(t')}{16\pi^2}\right] , \qquad I_i = \exp\left[\hat{b}\int_0^t dt' \frac{y_i^2(t')}{16\pi^2}\right] \qquad (i = e, \mu, \tau) . \tag{47}$$

To a good approximation we have:

$$I_K = \left(\frac{\mu}{\mu_0}\right)^{\frac{k}{16\pi^2}} \qquad I_e = I_\mu = 1 . \tag{48}$$

We evaluate $I_\tau$ by using eq. (45) and we get:

$$I_\tau = \left[\frac{(\hat{a}+1)}{u}y_\tau^2(\mu_0)\left(1 - \left(\frac{\mu}{\mu_0}\right)^{\frac{u}{8\pi^2}}\right) + 1\right]^{-\frac{\hat{b}}{2(\hat{a}+1)}} . \tag{49}$$

We can glue these solutions at the supersymmetry breaking scale $m_{SUSY}$ and cover the whole energy interval between the large scale $\Lambda$ and the electroweak scale, which for definiteness

we take as the $Z$ boson mass $m_Z$. The boundary values of $(\mathcal{Y}_e)_{MSSM}$ and $(\mathcal{W}_{ij})_{MSSM}$ at the scale $\Lambda$ are functions of our input parameters $a, b, c, \tau, \varphi_i$. Their boundary values at the scale $m_{SUSY}$ are obtained by requiring the continuity of the physical quantities, i.e. the charged lepton masses and the neutrino masses, see eq. (37):

$$
\begin{aligned}
(\mathcal{Y}_e)_{MSSM}(m_{SUSY})\cos\beta &= (\mathcal{Y}_e)_{SM}(m_{SUSY}) \\
(\mathcal{W}_{ij})_{MSSM}(m_{SUSY})\sin^2\beta &= (\mathcal{W}_{ij})_{SM}(m_{SUSY}) \quad .
\end{aligned}
\tag{50}
$$

The final values of $(\mathcal{Y}_e)_{SM}$ and $(\mathcal{W}_{ij})_{SM}$ at the scale $m_Z$ depend on the input parameters and on $\Lambda$, $m_{SUSY}$ and $\tan\beta$. We used as inputs: $g_Y = 0.34$, $g_2 = 0.64$, $\lambda = 0.13$. We kept as only non-vanishing quark couplings

$$
(y_t)_{SM} = \frac{\sqrt{2}m_t}{v} \quad , \qquad (y_t)_{MSSM} = \frac{\sqrt{2}m_t}{v \, \sin\beta} \quad , \qquad (y_b)_{MSSM} = \frac{\sqrt{2}m_b}{v \, \cos\beta} \quad ,
\tag{51}
$$

taking $v = 246$ GeV, $m_t = 163$ GeV, $m_b = 4.2$ GeV. From $m_Z$ down to lower energies the charged lepton masses still run through the electromagnetic interactions [63]. We account for this effect by directly fitting the values of the charged lepton Yukawa couplings at $m_Z$, as derived from [52, 66].

**Neutrino masses from the seesaw mechanism**

When the theory includes right-handed neutrinos we should also include running effects form the cut-off scale $\Lambda_{UV}$ down to the lightest heavy neutrino mass $M_1$, which we identify with the scale $\Lambda$ of our previous discussion [67]. At the scale $\Lambda_{UV}$ the leptonic sector is described by:

$$
\mathcal{L}_{\texttt{lept}} = -E^c \mathcal{Y}_e H_d L - N^c \mathcal{Y}_\nu H_u L - \frac{1}{2} N^c M N^c + h.c.
\tag{52}
$$

At the scale $\Lambda \equiv M_1$ we integrate out the heavy neutrinos and we go back to eq. (36), SUSY case, with $\overline{\mathcal{W}} = \overline{\mathcal{W}}(\Lambda)$, where we defined:

$$
\overline{\mathcal{W}}(\mu) \equiv \frac{\mathcal{W}(\mu)}{\Lambda} = \frac{1}{2} (\mathcal{Y}_\nu^T M^{-1} \mathcal{Y}_\nu)(\mu) \quad .
\tag{53}
$$

We are interested in the running of the quantities $\mathcal{Y}_e$, $\mathcal{Y}_\nu$ and $\overline{\mathcal{W}}$. In the one-loop approximation and neglecting threshold effects from $\Lambda_{UV}$ down to $M_1$, we have:

$$
\begin{aligned}
16\pi^2 \frac{d\mathcal{Y}_e}{dt} &= \mathcal{Y}_e \big[ 3\,\mathcal{Y}_e^\dagger \mathcal{Y}_e + \mathcal{Y}_\nu^\dagger \mathcal{Y}_\nu + \text{tr}(\mathcal{Y}_e^\dagger \mathcal{Y}_e) + u \big] \\
16\pi^2 \frac{d\mathcal{Y}_\nu}{dt} &= \mathcal{Y}_\nu \big[ 3\,\mathcal{Y}_\nu^\dagger \mathcal{Y}_\nu + \mathcal{Y}_e^\dagger \mathcal{Y}_e + \text{tr}(\mathcal{Y}_\nu^\dagger \mathcal{Y}_\nu) + w \big] \\
16\pi^2 \frac{d\overline{\mathcal{W}}}{dt} &= (k + 2\,\text{tr}(\mathcal{Y}_\nu^\dagger \mathcal{Y}_\nu))\,\overline{\mathcal{W}} + \big( \overline{\mathcal{W}}\,(\mathcal{Y}_e^\dagger \mathcal{Y}_e + \mathcal{Y}_\nu^\dagger \mathcal{Y}_\nu) + (\mathcal{Y}_e^\dagger \mathcal{Y}_e + \mathcal{Y}_\nu^\dagger \mathcal{Y}_\nu)^T\,\overline{\mathcal{W}} \big) \quad ,
\end{aligned}
\tag{54}
$$

where

$$
\begin{aligned}
u &= -3(g_Y^2 + g_2^2) + 3\,\text{tr}(\mathcal{Y}_d^\dagger \mathcal{Y}_d) \\
w &= -g_Y^2 - 3g_2^2 + 3\,\text{tr}(\mathcal{Y}_u^\dagger \mathcal{Y}_u) \\
k &= -2g_Y^2 - 6g_2^2 + 6\,\text{tr}(\mathcal{Y}_u^\dagger \mathcal{Y}_u) \quad .
\end{aligned}
\tag{55}
$$

In order to have analytical solutions to these equations we exploit the fact that at the boundary scale $\Lambda_{UV}$ the model under discussion predicts:

$$
(\mathcal{Y}_\nu^\dagger \mathcal{Y}_\nu) = y_0^2 \mathbb{1} \quad ,
\tag{56}
$$

and we discuss two representative cases.

**Case 1:** $\Lambda = 10^{15}$ **GeV**

In this case the eigenvalues of $\mathcal{Y}_\nu^\dagger \mathcal{Y}_\nu$ are of order one, in order to have neutrino masses at least of order $\sqrt{\Delta m_{atm}^2}$. The largest charged lepton Yukawa coupling, $y_\tau^2$, is of order $10^{-2}(\tan\beta/10)^2$, much smaller than one for $\tan\beta \leq 30$. To a good approximation we can drop all terms proportional to $\mathcal{Y}_e^\dagger \mathcal{Y}_e$ in the right-hand side of eqs. (54). The solution of these equations is a simple rescaling of the matrices $\mathcal{Y}_e$, $\mathcal{Y}_\nu^\dagger \mathcal{Y}_\nu$ and $\overline{\mathcal{W}}$:

$$
\begin{aligned}
(\mathcal{Y}_\nu^\dagger \mathcal{Y}_\nu)(\mu) &= \frac{\mathbb{1}}{\dfrac{6y_0^2 + w}{y_0^2 \, w} \left(\dfrac{\Lambda_{UV}}{\mu}\right)^{\frac{w}{8\pi^2}} - \dfrac{6}{w}} \equiv y^2(\mu)\mathbb{1} \\
\mathcal{Y}_e(\mu) &= \mathcal{Y}_e(\Lambda_{UV}) \exp\left[\int_0^t dt' \frac{(y^2(t')+u)}{16\pi^2}\right] \qquad\qquad (57) \\
\overline{\mathcal{W}}(\mu) &= \overline{\mathcal{W}}(\Lambda_{UV}) \exp\left[\int_0^t dt' \frac{(8y^2(t')+k)}{16\pi^2}\right] \qquad t = \log\mu/\Lambda_{UV} \,. \quad (58)
\end{aligned}
$$

The goodness of this approximation also relies on the fact that the interval from $\Lambda_{UV}$ down to $\Lambda$ is too short for the neglected effects to become relevant.

**Case 2:** $\Lambda = 10^{11}$ **GeV**

In this case the eigenvalues of $\mathcal{Y}_\nu^\dagger \mathcal{Y}_\nu$ are of order $10^{-4}$, smaller than $y_\tau^2$. To a good approximation we can drop all terms proportional to $\mathcal{Y}_\nu^\dagger \mathcal{Y}_\nu$ in the right-hand side of eqs. (54). The equations for $\mathcal{Y}_e$ and $\overline{\mathcal{W}}$ become identical to the ones discussed before in the range of energies below $\Lambda$ and the solutions (44-47) apply with $\hat{a} = 3$ and $\hat{b} = 1$.

When $\Lambda$ is well inside the range $(10^{11} \div 10^{15})$ GeV, we should solve the equations numerically.

# E  Fit to Model 2 with RGE Effects

Table 11: Full results of the fits as a function of $\tan\beta$ and $m_{SUSY}$ for Model 1.

| $m_{SUSY} = 10^4\,\mathrm{GeV}$ | | $\tan\beta = 2.5$ | $\tan\beta = 10$ | $\tan\beta = 15$ |
|---|---|---|---|---|
| | Parameters | $\tau = 0.0117 + 0.9947i,$ $\varphi_3 = -0.086,$ $a = 6.110670 \times 10^{-6},$ $b = 2.176072 \times 10^{-2},$ $c = 1.284381 \times 10^{-3}.$ | $\tau = 0.0117 + 0.9946i,$ $\varphi_3 = -0.086,$ $a = 2.344023 \times 10^{-5},$ $b = 8.382802 \times 10^{-2},$ $c = 4.926824 \times 10^{-3}.$ | $\tau = 0.0120 + 0.9946i,$ $\varphi_3 = -0.088,$ $a = 3.638217 \times 10^{-5},$ $b = 1.308358 \times 10^{-1},$ $c = 7.645341 \times 10^{-3}.$ |
| | Observables | $r = 0.0302,\quad \frac{m_3}{m_2} = 0.015,$ $s^2_{12} = 0.304,\quad s^2_{13} = 0.0217,$ $s^2_{23} = 0.578,\quad \frac{\delta}{\pi} = 1.530,$ $\frac{\alpha_{21}}{\pi} = 0.137,\quad \frac{\alpha_{31}}{\pi} = 1.731,$ $y_e = 2.794745 \times 10^{-5},$ $y_\mu = 5.899863 \times 10^{-4},$ $y_\tau = 1.002950 \times 10^{-2}.$ | $r = 0.0292,\quad \frac{m_3}{m_2} = 0.015,$ $s^2_{12} = 0.345,\quad s^2_{13} = 0.0217,$ $s^2_{23} = 0.577,\quad \frac{\delta}{\pi} = 1.523,$ $\frac{\alpha_{21}}{\pi} = 0.131,\quad \frac{\alpha_{31}}{\pi} = 1.724,$ $y_e = 2.794745 \times 10^{-5},$ $y_\mu = 5.899863 \times 10^{-4},$ $y_\tau = 1.002950 \times 10^{-2}.$ | $r = 0.0288,\quad \frac{m_3}{m_2} = 0.015,$ $s^2_{12} = 0.418,\quad s^2_{13} = 0.0212,$ $s^2_{23} = 0.574,\quad \frac{\delta}{\pi} = 1.513,$ $\frac{\alpha_{21}}{\pi} = 0.125,\quad \frac{\alpha_{31}}{\pi} = 1.708,$ $y_e = 2.794745 \times 10^{-5},$ $y_\mu = 5.899863 \times 10^{-4},$ $y_\tau = 1.002949 \times 10^{-2}.$ |
| | $\chi^2$ | 0.4 | 12.2 | 82.0 |

Table 11 Continued: Full results of the fits as a function of $\tan\beta$ and $m_{SUSY}$ for Model 1.

| $m_{SUSY} = 10^8 \text{GeV}$ | | $\tan\beta = 2.5$ | $\tan\beta = 10$ | $\tan\beta = 15$ |
|---|---|---|---|---|
| | Parameters | $\tau = 0.0117 + 0.9947i,$ <br> $\varphi_3 = -0.086,$ <br> $a = 7.212383 \times 10^{-6},$ <br> $b = 2.568250 \times 10^{-2},$ <br> $c = 1.515947 \times 10^{-3}.$ | $\tau = 0.0117 + 0.9946i,$ <br> $\varphi_3 = -0.086,$ <br> $a = 2.739916 \times 10^{-5},$ <br> $b = 9.788823 \times 10^{-2},$ <br> $c = 5.758937 \times 10^{-3}.$ | $\tau = 0.0118 + 0.9946i,$ <br> $\varphi_3 = -0.087,$ <br> $a = 4.197150 \times 10^{-5},$ <br> $b = 1.506051 \times 10^{-1},$ <br> $c = 8.820872 \times 10^{-3}.$ |
| | Observables | $r = 0.0302, \quad \frac{m_3}{m_2} = 0.015, \quad s_{13}^2 = 0.0217,$ <br> $s_{12}^2 = 0.303, \quad \frac{\delta}{\pi} = 1.530,$ <br> $s_{23}^2 = 0.578, \quad \frac{\alpha_{31}}{\pi} = 1.733,$ <br> $\frac{\alpha_{21}}{\pi} = 0.137,$ <br> $y_e = 2.794745 \times 10^{-5},$ <br> $y_\mu = 5.899863 \times 10^{-4},$ <br> $y_\tau = 1.002950 \times 10^{-2}.$ | $r = 0.0294, \quad \frac{m_3}{m_2} = 0.015, \quad s_{13}^2 = 0.0217,$ <br> $s_{12}^2 = 0.335, \quad \frac{\delta}{\pi} = 1.525,$ <br> $s_{23}^2 = 0.577, \quad \frac{\alpha_{31}}{\pi} = 1.726,$ <br> $\frac{\alpha_{21}}{\pi} = 0.132,$ <br> $y_e = 2.794745 \times 10^{-5},$ <br> $y_\mu = 5.899863 \times 10^{-4},$ <br> $y_\tau = 1.002950 \times 10^{-2}.$ | $r = 0.0286, \quad \frac{m_3}{m_2} = 0.015, \quad s_{13}^2 = 0.0214,$ <br> $s_{12}^2 = 0.389, \quad \frac{\delta}{\pi} = 1.517,$ <br> $s_{23}^2 = 0.575, \quad \frac{\alpha_{31}}{\pi} = 1.715,$ <br> $\frac{\alpha_{21}}{\pi} = 0.127,$ <br> $y_e = 2.794745 \times 10^{-5},$ <br> $y_\mu = 5.899863 \times 10^{-4},$ <br> $y_\tau = 1.002949 \times 10^{-2}.$ |
| | $\chi^2$ | 0.4 | 7.0 | 47.7 |

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
