# Peer review of "Modular Invariance Faces Precision Neutrino Data"

_SciPost Physics, doi:SciPost Phys. 5, 042 (2018)_

## Round 2 · Referee Report · George Leontaris (Referee 1) · 2018-9-15

Strengths

1)The use of Modular invariance is an interesting alternative approach to mass textures
2) Good predictions on the neutrino sector

Weaknesses

1) Arguments about charged lepton mass matrix are rather weak
2) Implications of modular invariance on the quark sector not discussed.

Report

In this work the authors pursue an approach based on modular invariance
to determine the lepton mass matrices and give predictions for neutrino data.
In their analysis the neutrino masses are generated either from a Weinberg operator or from a see-saw mechanism. In both cases, modular symmetry puts strong constraints and reduces the number of arbitrary parameters. RGEs and SUSY effects are also taken into account.

This work is an interesting improvement and continuation of past effords in
this particular subject. It is also well motivated since modular invariance
appears also in string theories.

In general the paper is well written and the results agree with the existing data. However, I find the discussion of section 3 (above equation (18)), regarding the issue of the reactor angle $\theta_{13}$ and the charged lepton
mass matrix, somewhat confusing and not very convincing.
At least in the way it is presented, it is also not clear what exactly the authors are using from ref[20].
Also, some expectations for the quark sector should be discussed.
In section 5 the author refer to "approximate symmetry" for SUSY. I think that they mean broken supersymmetry.

There are also a few minor flaws in the text. For example "by a phase transformations" (page 2), Kahler instead of K\"ahler, a missing fullstop, "loosing generality" instead of the math-jargon "without loss of generality"
( page 8) .
I would recommend publication after the authors consider the above corrections and suggestions.

Requested changes

1) Improvement and clarification of discussion in section 3, above eq (18)
2) Short discussion for the quark sector
3) Correction on language mistakes

---

## Round 2 · Referee Report · Anonymous (Referee 2) · 2018-9-23

Strengths

1- Original aproach to lepton masses and mixings patterns
2- The comparison with the data is excellent
3- A relatively small number of parameters used, leading to several predictions
4- Clarity of the arguments, in a subject that is rather technical

Weaknesses

It would be interesting to find the vacuum structure by a dynamical minimization starting from a microscopic model for the flavons and the modulus field.

Report

Excellent paper using an original approach, using techniques from field theory, string theory and neutrino physics. Impressive amount of work with an excellent output in comparing with data.
I am very happy to propose the paper for publication in the current form.
Just a question for my curiosity: in eq. 3, Kahler transformations require that the superpotential transforms with $e^{-f}$. This term becomes one in the global SUSY limit $M_P \to \infty$, but is necessary in supergravity. Moreover, in the simplest examples of compactifications, superpotential Yukawas do not have zero modular weight. It seems to me that the invariance of the superpotential under modular transformations was imposed by hand. Unless the invariance under the discrete flavor symmetry $\Gamma_3$ is the key ingredient. Am I wrong ?

Requested changes

None

  • validity: top
  • significance: top
  • originality: top
  • clarity: top
  • formatting: perfect
  • grammar: excellent

Author:  Ferruccio Feruglio  on 2018-09-24  [id 321]

(in reply to Report 2 on 2018-09-23)
Category:
answer to question

First of all, thank you for your comments.

Concerning the case of local supersymmetry you are perfectly
right. The superpotential is not expected to be modular invariant. For instance, with the `minimal'
choice of Kahler potential done in our paper, namely $K=-h log(-i \tau+i\bar\tau)+...$,
the superpotential should transform with weight $-h$, that is $w\to (c \tau+d)^{-h} w$.
However this fact can be easily incorporated in our construction by suitably shifting the weight
assigned to the matter supermultiplets, our eq. (2). In the bottom-up approach we are considering,
the choice of the weights for the matter supermultiplets is part of the freedom of the model.
Also in a sugra context, with an appropriate choice of the weights, we can reproduce the same Yukawa
couplings of the rigid case, while allowing the superpotential $w$ to transform non trivially
under modular transformations as requested.

Please do not hesitate to ask further questions if this explanation is not sufficiently clear.

---

## Round 3 · Referee Report · George Leontaris (Referee 1) · 2018-9-27

Report

The present version of the paper clarifies all the points required in my report,
in particular those regarding theta_13 angle and quark sector.
In my opinion it is now suitable for publication.

---

## Round 3 · Referee Report · Anonymous (Referee 2) · 2018-9-28

Report

The authors answered satisfactory to my question. I am therefore pleased to propose the paper for publication in the current form.

---

## Round 3 · Author Response

Dear Editor and Referees, we have modified our text following your suggestions. In particular

  • In section 3, second paragraph, we have hopefully better clarified what results of reference [20] we are using and the role played by the reactor angle in our analysis. This part has been completely rewritten aiming to a more transparent presentation.
  • In section 3, at the end, we have added a paragraph where we briefly mention the implications of modular invariance for the quark sector, as requested.
  • Finally we have corrected the grammatical errors/typos.

We hope that the present version fulfills your requirements.

                       Kind regards,           Juan Carlos Criado and Ferruccio Feruglio

---

## Round 3 · List of Changes

• In section 3, second paragraph, we have hopefully better clarified what results of reference [20] we are using and the role played by the reactor angle in our analysis. This part has been completely rewritten aiming to a more transparent presentation.
  • In section 3, at the end, we have added a paragraph where we briefly mention the implications of modular invariance for the quark sector, as requested.
  • Finally we have corrected the grammatical errors/typos.

---

## Editorial Decision

published